# Analysis of 13,312 benthic invertebrate samples from German streams reveals minor deviations in ecological status class between abundance and presence/absence data

**Dominik Buchner**[1☯], **Arne J. Beermann**[1,2], **Alex Laini**[3], **Peter Rolauffs**[4], **Simon Vitecek**[5,6], **Daniel Hering**[2,4], **Florian Leese**[1,2☯]*

1 University of Duisburg-Essen, Aquatic Ecosystem Research, Essen, Germany, 2 Centre for Water and Environmental Research (ZWU), Essen, Germany, 3 University of Parma, Department of Chemistry, Life Sciences and Environmental Sustainability, Parma, Italy, 4 University of Duisburg-Essen, Aquatic Ecology, Essen, Germany, 5 WasserCluster Lunz, Lunz am See, Austria, 6 University of Natural Resources Vienna, Wien, Austria

☯ These authors contributed equally to this work.
* florian.leese@uni-due.de

**Data Availability Statement:** Supporting Information S2–S4 Tables contain metrics results

## Abstract

Benthic invertebrates are the most commonly used organisms used to assess ecological status as required by the EU Water Framework Directive (WFD). For WFD-compliant assessments, benthic invertebrate communities are sampled, identified and counted. Taxa × abundance matrices are used to calculate indices and the resulting scores are compared to reference values to determine the ecological status class. DNA-based tools, such as DNA metabarcoding, provide a new and precise method for species identification but cannot deliver robust abundance data. To evaluate the applicability of DNA-based tools to ecological status assessment, we evaluated whether the results derived from presence/absence data are comparable to those derived from abundance data. We analysed benthic invertebrate community data obtained from 13,312 WFD assessments of German streams. Broken down to 30 official stream types, we compared assessment results based on abundance and presence/absence data for the assessment modules "organic pollution" (i.e., the saprobic index) and "general degradation" (a multimetric index) as well as their underlying metrics.

In 76.6% of cases, the ecological status class did not change after transforming abundance data to presence/absence data. In 12% of cases, the status class was reduced by one (e.g., from good to moderate), and in 11.2% of cases, the class increased by one. In only 0.2% of cases, the status shifted by two classes. Systematic stream type-specific deviations were found and differences between abundance and presence/absence data were most prominent for stream types where abundance information contributed directly to one or several metrics of the general degradation module. For a single stream type, these deviations led to a systematic shift in status from 'good' to 'moderate' (n = 201; with only n = 3 increasing). The systematic decrease in scores was observed, even when considering simulated confidence intervals for abundance data. Our analysis suggests that presence/

and correlation analyses based on 13,312 individual taxa lists. These raw data (taxa lists for the individual stream sites) cannot be shared publicly because they are owned by the individual federal stated. All raw data were requested from and are available according to the Environmental Information Act (Umweltinformationsgesetzt, UIG, 14.2.2005) from the individual federal states/institutions/persons as stated below. The authors did not receive special access privileges to the data: Baden-Wurttemberg: LUBW Landesanstalt für Umwelt, Messungen und Naturschutz Baden-Württemberg; Andreas Hoppe - Andreas. Hoppe@lubw.bwl.de; Bavaria: Bayerisches Landesamt für Umwelt; Folker Fischer-poststelle@lfu.bayern.de; Hesse: Hessisches Landesamt für Naturschutz, Umwelt und Geologie; Elisabeth Schlag - Elisabeth.Schlag@hlnug.hessen. de; Mecklenburg-West Pomerania: Landesamt für Umwelt, Naturschutz und Geologie Mecklenburg-Vorpommern Abteilung Geologie, Wasser und Boden; Andre Steinhaeuser - andre. steinhaeuser@lung.mv-regierung.de; Lower Saxony: Niedersaechsischer Landesbetrieb für Wasserwirtschaft, Küsten- und Naturschutz; Eva Bellack - Eva.Bellack@nlwkn-hi.niedersachsen.de North Rhine-Westphalia: Jochen Lacombe (Landesamt fuer Umwelt-, Natur- und Verbraucherschutz NRW, LANUV) - Jochen. Lacombe@lanuv.nrw.de; Saxony: Sächsisches Landesamt für Umwelt, Landwirtscahft und Geologie, Abteilung Wasser, Boden und Wertstoffe und Betriebsgesellschaft für Umwelt und Landwirtschaft; Antje Mickel - Antje.Mickel@smul. sachsen.de; Saxony-Anhalt: Landesbetrieb für Hochwasserschutz und Wasserwirtschaft Sachsen-Anhalt; Data are freely accessible or can be requested via Martina Jährling - http://gldweb. dhi-wasy.com/gld-portal/; Schleswig Holstein: Landesamt für Landwirtschaft, Umwelt und ländliche Räume SH; Annegret Holm - Annegret. Holm@llur.landsh.de.

**Funding:** The authors received no specific funding for this work.

**Competing interests:** The authors have declared that no competing interests exist.

absence data can yield similar assessment results to those for abundance-based data, despite type-specific deviations. For most metrics, it should be possible to intercalibrate the two data types without substantial efforts. Thus, benthic invertebrate taxon lists generated by standardised DNA-based methods should be further considered as a complementary approach.

## Introduction

Status assessment of freshwater ecosystems is frequently performed with biological indicators. They are of particular importance in Europe, where the Water Framework Directive (Directive 2000/60/EC; WFD) requires EU member states to achieve 'good ecological status' for all water bodies by 2027, defined as a 'slight deviation from undisturbed conditions.' Ecological status is determined based on biological quality elements (BQEs), i.e., organism groups that reflect aquatic ecosystem integrity by responding to various pressures rather than the intensity of a single pressure. For rivers, the aquatic flora (phytobenthos and macrophytes), fish and, most frequently, benthic invertebrates are monitored. Assessment methods are defined for individual stream types (ST), defined by their sizes, ecoregions, or catchment geology, differing in their biota and resilience to stress [1]. The assessment procedure typically involves the standardised sampling of the BQE community, enumeration of taxa and estimating taxon abundance. Based on the resulting taxa lists, metrics are calculated and compared to reference values derived from undisturbed reference sites or through a modelling approach [2]. The resulting score is then translated into an ecological status class (ESC) of high, good, moderate, poor, or bad.

In compliance with the legal requirement of the WFD, all EU member states have developed nationwide assessment tools to monitor ecological status. While the same principles are applied across the entire EU, details of the assessment systems differ among member states. These differences are rooted in local monitoring traditions, different pressures affecting the water bodies, and biogeography. Assessment systems of different countries have been intercalibrated to enable comparisons of results [3].

The majority of national assessment systems rely on multimetric indices, i.e., combinations of quantitative or qualitative descriptors of a certain aspect of an ecosystem based on the taxon list, such as the number of taxa, share of sensitive species, or abundance of an indicator group [1]. Though the WFD requires the use of taxon abundances, the individual metrics do not necessarily use raw abundance data; many are based on taxon number, presence/absence of taxa, or abundance classes.

The surge in DNA-based community assessment [4,5] has raised questions about whether DNA-based identification can supersede morphological identification procedures and be used for assessment systems under the WFD. The DNA-based characterisation of biotic communities can uncover diversity patterns at high taxonomic resolution [6,7]. However, it is generally appreciated that PCR-based approaches, such as metabarcoding, cannot deliver reliable absolute abundance data for Metazoa [8,9], although strong and positive correlations between read number and biomass are sometimes found (e.g. [7,10,11]). To a lesser degree, the same has been found for single-celled organisms [12,13]. Presence/absence data alone, as inferred reliably using DNA-based tools, are incompatible with WFD requirements. However, given the benefits of molecular biomonitoring tools in terms of taxonomic, spatial and temporal resolution, it seems worthwhile exploring the use of presence/absence data to infer established

indices. While we are well aware that these tools were not designed for presence/absence data, we think that the implementation of biomonitoring 2.0 could be achieved by two different strategies [14]. First, molecular data could be used to simply replace the classical taxa × abundance matrices with taxa × presence/absence matrices obtained by the molecular characterisation of communities. Second, an entirely new set of metrics and indices based on molecular data could be trained and calibrated to determine pressure gradients and human impact. While rebuilding the entire biomonitoring toolset, a formidable challenge, is likely the best way to deliver more specific assessments at higher resolution, its implementation will be costly and may thus be less attractive for policy-makers, requiring fundamental changes in legislation. Therefore, the ability to simply adapt existing tools to presence/absence data (which can be obtained quickly and reliably using molecular tools, such as metabarcoding) should be gauged [15,16]. Some studies of freshwater invertebrates [17,18] as well as case studies of marine invertebrates [19] have suggested a general coherence of assessment results between the two data types. However, systematic and large-scale studies directly comparing abundance and presence/absence data are scarce.

We performed the first nationwide comparison of river ESCs calculated with abundance data and with presence/absence data. Specifically, we hypothesise that (i) there is general congruence between metric or module results based on abundance and presence/absence data and (ii) the proportion of metrics that use raw abundance data determines how well presence/absence-based assessment results (i.e., the assignment of ESCs) correspond with the results of an abundance-based approach. We used a large dataset including over 13,000 samples from German WFD compliance monitoring using the PERLODES system. PERLODES includes metrics based on presence/absence, raw abundance and abundance classes, and is thus representative of a variety of approaches developed for BQEs also in other countries [1]. In addition to hypothesis testing, we describe type- and class-specific mismatches between assessment results.

## Materials and methods

### German stream assessment methodology

The German assessment system PERLODES using benthic invertebrates [20,21] was selected for our analysis. It is based on a national river typology with 30 river types. For each river type, the assessment system uses two modules, each of which provides an ESC classification: organic pollution module (OPM; based on a single metric, the saprobic index, SI) and general degradation module (GDM; integrating three to five metrics, depending on stream type (ST), see S1 Table). The GDM reflects the effects of various pressures, particularly habitat degradation, on the benthic invertebrate fauna. A core element of this module is the German fauna index (GFI), which relies on the abundance of specialist indicator taxa that mainly occur in near-natural or degraded habitats. It always accounts for 50% of the module's result and is accompanied by two to four additional metrics, one of which, in most cases, is the proportion of Ephemeroptera, Plecoptera and Trichoptera specimens (EPT [%]). A third module, acidification (AM), was only applied to two STs, where the recovery process from previous acidification is ongoing. The final ESC is defined by the worst assessment result based on the OPM and the GDM and receives a classifier from high (ESC1) to poor (ESC5).

The metrics of the PERLODES system use different components of the underlying taxa × abundance list. While some metrics (e.g., the number of Trichoptera species) use (i) raw taxa numbers without abundance, others (e.g., the share of taxa preferring the hyporhithral zone) use (ii) raw taxa × abundance matrices, and others (e.g., the SI) use (iii) taxon × abundance

class matrices (see S1 Table; for details on metric and moduls calculation as well as abundance classes consult [22]).

## Data source and permissions

Access to biological monitoring data was granted by nine of the largest German federal states (Bavaria, Baden-Wuerttemberg, Hesse, Lower Saxony, North Rhine-Westphalia, Saxony, Saxony-Anhalt and Schleswig-Holstein) through the LAWA (German expert Working Group on water issues of the Federal States). The total data set contained 13,401 samples obtained from monitoring sites in 1985–2013, covering 30 STs [23].

## Data processing

To compare assessment results based on abundance or presence/absence data, raw data were used untransformed as well as transformed to presence/absence data for analyses in ASTERICS v. 4.04 [22]. Abundances in the raw data were replaced by either 1 (presence) or 0 (absence) for this transformation. Of the 13,401 samples, 89 had to be excluded owing to errors in the calculation of indices using ASTERICS, resulting in a total of 13,312 taxon lists. From the ASTERICS output, the results for the three modules (OPM, GDM and [if relevant] AM) and the final ESC as well as all relevant metrics for each ST (S1 Table) were exported to a .csv file. Data were extracted from this output file using a custom python script (S1 File).

## Statistical analysis

**i) Correlation and slope analysis.** The correlations between abundance and presence/absence module results, individual metric results and final ESC were determined. Spearman's rank correlation coefficients were obtained using R [cor.test] (https://www.r-project.org) because data were not normally distributed. A linear regression analysis was performed to evaluate the relationship between abundance (*x*) against presence/absence (*y*) data to test for systematic deviations, as indicated by slopes deviating from 1 (see S1 Fig) as well as Spearman's ρ values.

**ii) ESC and metric deviations.** To quantify deviations between assessment results for the three modules and the final ESC from abundance data, a custom python script summarised cases that remained unchanged (0) or deviated by one, two, or three classes. This was performed for the entire dataset and for discrete ESCs to test if sites of a certain ESC were more prone to changes (e.g., due to differences in taxa numbers). We focussed on shifts between 'good' and 'moderate' ESCs, as systematic deviations from these classes would confound efforts to attain a 'good' ESC, as requested by the WFD.

**iii) Deviation analysis.** Deviations between abundance-based and presence/absence-based ESC assessments were predicted to be most severe in STs where the GDM largely relies on metrics that directly use abundance data (see S1 Table). To test this hypothesis, we estimated the contribution of abundance-based metrics to the assessment result and related this to the cumulative percentages of mismatches between abundance-based and presence/absence-based assessment results ($C_{Ab}$). As the GFI always contributes 50% to the GDM, the contribution of any other metric ($C_{metric}$) besides the Fauna Index was defined as follows:

$$C_{metric} = \frac{0.5}{\sum(individual\ metrics) - 1}$$

$C_{metric}$ had values of 12.5%, 16.7%, 25.0%, or 100% (type 10 and 20) depending on the number of metrics used for the calculation (S1 Table). We then multiplied the value with the

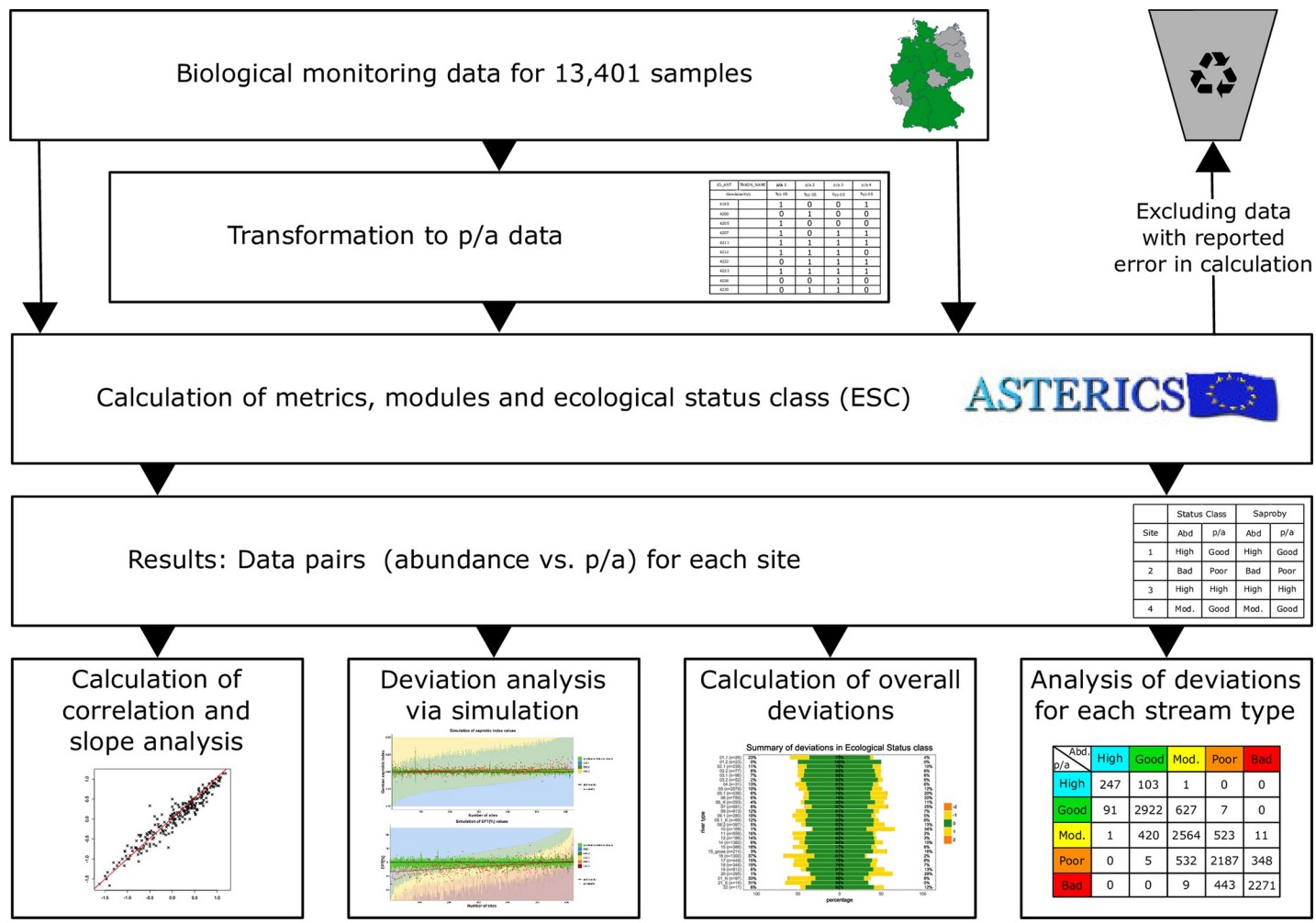

**Fig 1. Overview of the study workflow.**

number of metrics that actually relied on raw abundance data (e.g. two of the five metrics for ST16, i.e. Lit% and Pel%, see S1 Table) to obtain $C_{AB}$, which had values of 0%, 16,7% or 25%. Finally, we tested for a correlation between the magnitude of the deviation in the ESC based on abundance or presence/absence data and $C_{AB}$ per ST using Spearman's rank correlation.

The workflow for the analysis is summarised in Fig 1. All data are available on request.

## Simulation of taxa lists

Within-site patchiness of macroinvertebrate communities can bias abundance estimates inferred by using standard sampling protocols. To test if the deviations in ESC estimates based on abundance or presence/absence data could be the result of non-representative sampling, a simulation was performed accounting for uncertainties in abundance estimates: For a subset of sampling sites, we produced 1000 replicates each by drawing abundances from a zero-truncated Poisson distribution while retaining the original taxa list. The susceptibilities of SI, EPT [%] and GFI to within-site variability of abundances were thus tested in 627 sampling sites in which a change in ESC from 'good' to 'moderate' was observed in our previous analyses. For these, metrics were calculated using a custom python script, as ASTERICS cannot handle large

amounts of data. For each metric, a confidence interval spanning the 2.5[th] and 97.5[th] percentile was computed from the simulated data using another custom python script and functions from the *numpy* package. Results were plotted in R using the ggplot2 package.

## Results

### Overall congruence between abundance and presence/absence ESC estimates

Ecological status class (ESC) estimates inferred from presence/absence data and abundance data were congruent in 76.6% of all cases (10,191 of 13,312 comparisons; Fig 2, Table 1). Different ESCs were observed in 23.4% of all cases; 12% of presence/absence ESCs were one class lower than originally inferred (i.e., a shift towards a worse ecological status) and 11.2% of presence/absence ESCs were one class higher (i.e., a shift towards a better ecological status). Differences spanning more than one ESC were found in only 0.2% of cases, including 19 sites classified as worse and 15 sites classified as better by two ESCs.

We observed the strongest (>20%) bias towards a lower status class inferred from presence/absence data for stream types (ST)16 (small gravel-dominated lowland rivers; 37%, n = 1300), ST21_N (lake outflows, lowlands; 33%, n = 97), ST21_S (lake outflows, highlands; 31%, n = 16) and ST1.1 (small and mid-sized rivers of the Calcareous Alps; 23%, n = 26) (Fig 2). By contrast, STs most often (>20%) assigned to one or two status classes higher were ST10

**Fig 2. Deviations in ecological status class (ESC) estimates obtained from presence/absence data using ASTERICS for 13,312 samples from 30 German stream types (y-axis, numbers in parentheses = data points).** Green indicates the proportion of ESCs that remained identical, yellow indicates those that differed by ±1 and orange indicates those that differed by ±2 ESCs. Deviations in the negative (worse) direction are shown on the left and deviations towards a more positive assessment (better) are shown on the right side.

(very large gravel-dominated rivers; 34%, n = 169), ST20 (very large sand-dominated rivers; 29%, n = 295) and ST7 (small coarse substrate-dominated calcareous highland rivers; 25%, n = 681). The lowest overall deviations in ESC estimates were observed in ST1.2 (large rivers of the Calcareous Alps; 0% deviation, n = 23) and ST3.2 (mid-sized rivers in the Pleistocene sediments of the alpine foothills; 8% deviation, n = 52).

As predicted, ESC estimates in presence/absence-based and abundance-based assessments were highly congruent in all STs, as reflected by a significant correlation for all STs (p < 0.001; mean Spearman's ρ = 0.86, range 0.72–1, see S2 Table for details).

## Responses of individual modules

Similar to the results obtained for final ESCs, general degradation module (GDM) results based on presence/absence and abundance data were identical in 74.9% (n = 9956) of cases (Fig 3). GDM results decreased by one (-1) class in 13.1% (n = 1740) and by two classes (-2) in 0.1% of cases (n = 12), and increased by one (+1) and two (+2) classes in 11.8% (n = 1567) and 0.1% (n = 12) of all cases, respectively. Similarly, the highest deviations (>20%) in the GDM results were observed, with a single exception, in STs where ESC estimates showed the highest deviations (one and two status classes lower: ST16 (38%), ST21_S (38%), ST21_N (33%), ST1.1 (27%), ST17 (25%), ST9.1 (24%) and ST15 (22%); one and two status classes higher: ST10 (34%), ST20 (31%) and ST7 (26%)). The most congruent results were obtained for ST1.2 in which only 9% of all GDM results decreased by one class. GDM results were responsible for 95.06% and 95.57% of the observed shifts in ESC, i.e. of similarly great importance for both data sets.

Deviations in the organic pollution module (OPM) were moderate, with 91.7% congruent results (Fig 4). For ST1.1, ST1.2 and ST4, results were identical; the highest deviations in the OPM were observed in ST22 (marshland streams of the coastal plains; 18% of results differed).

Regression slopes describing relationships between abundance-based and presence/absence-based metric results for the two most relevant GDM metrics (i.e., GFI and EPT [%]; both used in 28 of 30 STs) generally deviated from 1, where a low ecological quality corresponded with higher scores (Fig 5A and 5B). Similarly, regression slopes describing relationships between abundance-based and presence/absence-based OPM results were, on average, less than 1 (mean = 0.92, min = 0.83 for ST3.2 and max = 1.1 for ST1.2, see Fig 6), with an average slope of 0.922.

## General patterns of metric congruence in relation to data characteristics

As expected, metrics that use presence/absence of taxa (e.g. number of Trichoptera taxa) were perfectly correlated when calculated with presence/absence data (Spearman's ρ = 1, S3 Table). Metrics that use abundance classes, such as the GFI, saprobic index, or rheo index, showed generally strong and significant correlations (mean Spearman's ρ = 0.93, range: 0.89–0.96) between both data types. We observed the weakest correlations for metrics that rely on raw abundance data, e.g., the relative proportion of individuals that prefer the epirhithral, metarhithral, or hyporhithral zones (mean Spearman's ρ = 0.60, range: 0.41–0.79).

We found a significant, positive correlation between the relative contribution of abundance data to GDM results and the magnitude of the deviation (Spearman's ρ = 0.52, p = 0.004, S2 Fig).

## Class boundary deviations

We found that presence/absence-based ESC estimates differed in 23% of the observed cases. Of major practical importance are shifts from ESC 2 ('good') to ESC 3 ('moderate', i.e. not

**Table 1. Deviation between presence/absence and abundance data for different assessment metrics.** The diagonal displays the numbers of sampling sites for which the results did not differ between the two data types.

| | | Presence/absence data | | | | |
|---|---|---|---|---|---|---|
| | Ecological status class | 1 | 2 | 3 | 4 | 5 |
| Abundance data | 1 | 247 | 103 | 1 | 0 | 0 |
| | 2 | 91 | 2922 | 627* | 7 | 0 |
| | 3 | 1 | 420 | 2564 | 523 | 11 |
| | 4 | 0 | 5 | 532 | 2187 | 348 |
| | 5 | 0 | 0 | 9 | 443 | 2271 |
| | | Presence/absence data | | | | |
| | Organic pollution | 1 | 2 | 3 | 4 | 5 |
| Abundance data | 1 | 821 | 147 | 0 | 0 | 0 |
| | 2 | 165 | 8758 | 307 | 0 | 0 |
| | 3 | 0 | 428 | 2597 | 5 | 0 |
| | 4 | 0 | 0 | 49 | 31 | 1 |
| | 5 | 0 | 0 | 0 | 2 | 1 |
| | Presence/absence data | | | | | |
| | General degradation | 1 | 2 | 3 | 4 | 5 |
| Abundance data | 1 | 698 | 263 | 3 | 0 | 0 |
| | 2 | 162 | 2295 | 616 | 6 | 0 |
| | 3 | 2 | 426 | 2521 | 517 | 7 |
| | 4 | 0 | 5 | 534 | 2175 | 344 |
| | 5 | 0 | 0 | 9 | 445 | 2267 |

* indicates the number of stream assessments that would fall outside the threshold criterion (i.e. at least ESC = 2) after transformation. The high deviation is due to stream type 16.

meeting the requirements of the WFD). We found that 627 'good' ESCs were classified as 'moderate', and 420 cases originally classified as 'moderate' were recovered as 'good' when using presence/absence data (Table 1). The highest proportion of cases showing a shift from 'good' to 'moderate' belonged to ST16 (small gravel-dominated lowland rivers) with 201 mis-classified cases (S4 Table). We observed frequent misclassifications of 'bad' ESCs as 'poor' (n = 445) as well as vice versa (n = 348).

Detailed analyses based on simulated abundance data for the 627 cases where presence/absence data suggested 'moderate' instead of the original 'good' ESC indicate that this misclassification is due to the GDM but not the OPM (S3 Fig). There was also substantial overlap in the simulation-derived confidence intervals of assessment results and assessment results based on presence/absence-based data, despite using a very conservative Poisson-distributed simulation approach.

## Discussion

### General deviation patterns

In agreement with hypothesis (i), our analysis revealed strong congruence between abundance and presence/absence data for most modules and metrics. The slope of the regression, however, was <1.0 for all stream types (STs) with abundant sampling data, i.e., n > 100 data points. This indicates that the metric value spectrum underlying ecological status class (ESC) calculation is narrower, i.e., biased towards intermediate values, when using presence/absence data. Deviation of ESC was also substantially greater when metrics in the general degradation module (GDM) relied to a larger degree on raw abundance data (S2 Fig).

# Summary of deviations in module general degradation

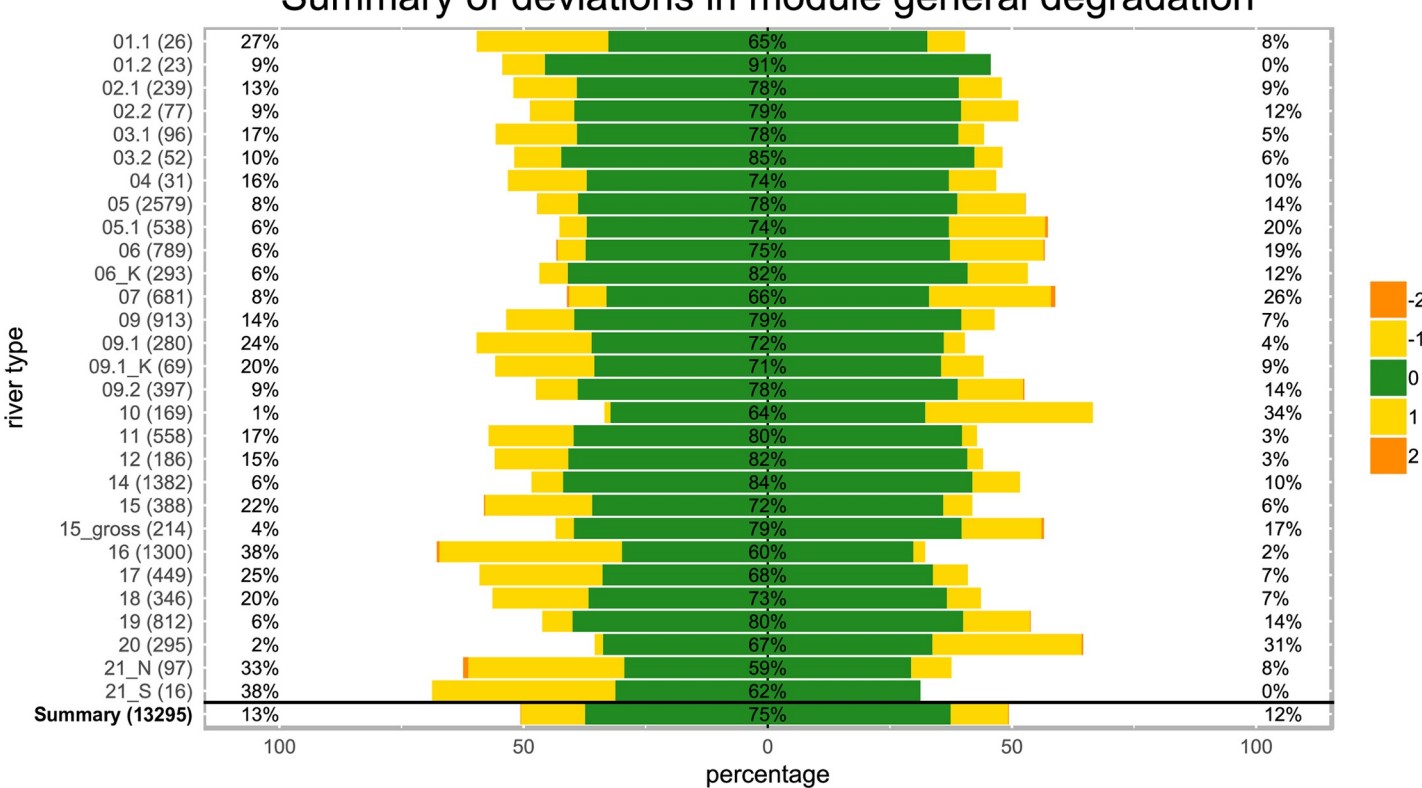

**Fig 3. Deviations in general degradation module (GDM) results calculated from presence/absence data using ASTERICS for 13,295 samples from 30 German stream types (y-axis, numbers in parentheses = data points).** Green indicates assessment values that remained identical, yellow indicates those that differed by ±1 and orange indicates those that differed by ±2 classes. Deviations in the negative (worse) direction are shown on the left and deviations towards a more positive assessment (better) are shown on the right side.

Over 75% of the ESCs remained unchanged after the transformation of abundance data to presence/absence data, while less than 25% of cases were classified as one ESC lower or higher. This shift can typically be explained by changes in the GDM while the organic pollution module (OPM) is more robust. Also, our simulation experiments demonstrate that the GDM is more sensitive to variation in abundance data: In about 95% of cases the ESC is determined by the GDM. In particular, the systematic shift from good to moderate (ESC2 to ESC3) has immense policy implications because the latter indicates a failure to comply with the WFD requirements. Therefore, we analysed this class boundary in detail and simulated Poisson-distributed confidence intervals for abundance data. While about 50% of the presence/absence data fell within the confidence interval, a high percentage of data points were in a lower category (S3 Fig). The observed deviation suggests that there are, on average, slightly greater differences between abundance-based and presence/absence-based assessments than the difference in results obtained by two independent investigators performing morphological identification (about 16%[24]).

## Specific deviation patterns

We found a notable bias when comparing presence/absence and abundance data for ST16 (small gravel-dominated lowland rivers), for which 1,300 data points were available. This is entirely due to the GDM, which is composed of five metrics for ST16 (S3 Table). While strong

## Summary of deviations in module organic pollution

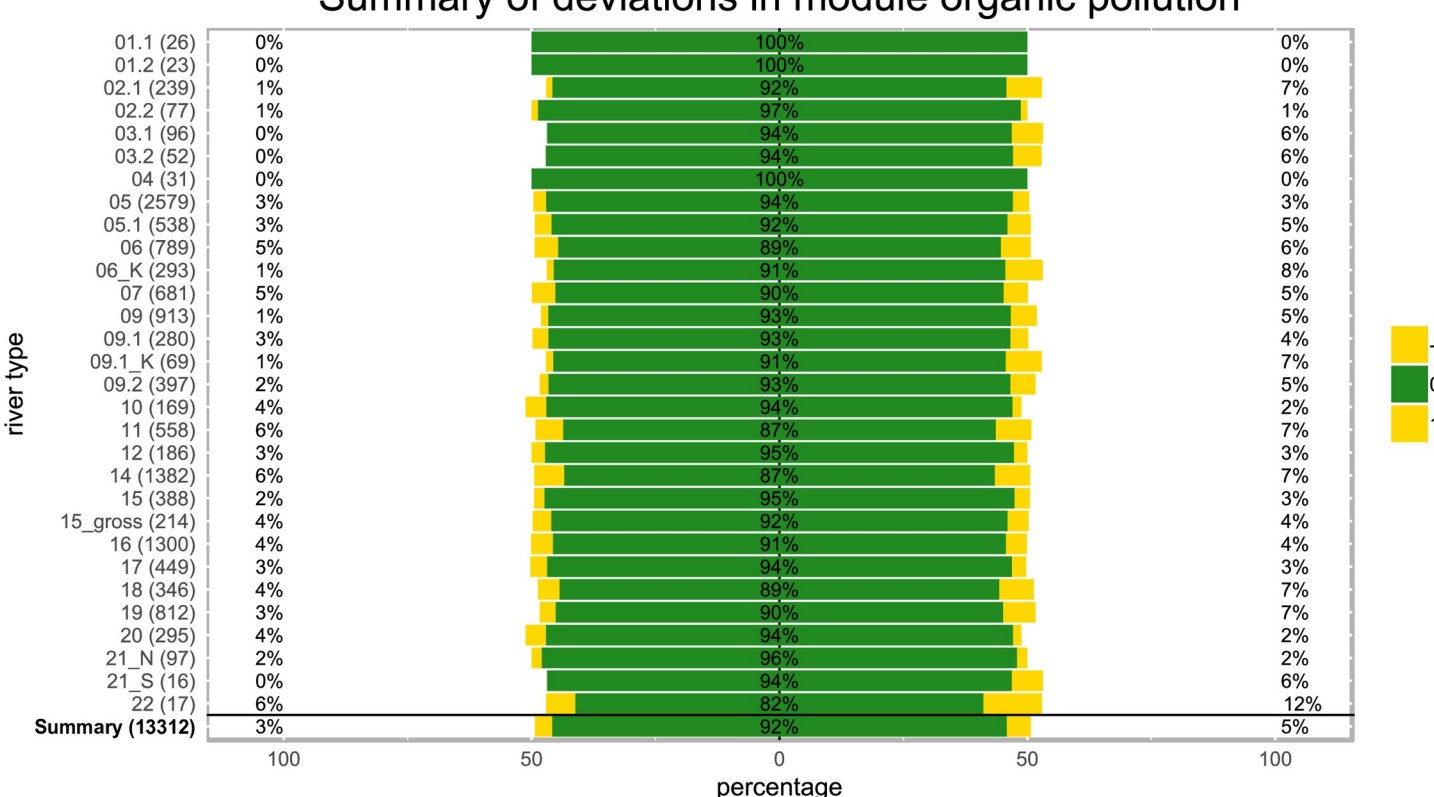

**Fig 4. Deviations in organic pollution module results calculated with presence/absence data using ASTERICS for 13,312 samples from 30 different German stream types (y-axis, numbers in parentheses = data points).** Green indicates assessment values that remained identical, yellow indicates values that differed by ±1 and orange indicates value that differed by ±2 classes. Deviations in the negative (worse) direction are shown on the left and deviations towards a more positive assessment (better) are shown on the right side.

positive correlations between data types were found for the GFI, the number of Trichoptera taxa and EPT [%], the results for proportions of pelal-inhabiting (mud-dwelling) and littoral taxa (typical for shallow areas in lentic waters) were poorly correlated ($\rho$ = 0.41 and 0.56). This can be explained by the large number of mud-dwelling and littoral taxa, albeit with low abundances, and the fact that both metrics use raw abundance data (S3 Table, red shading). As presence/absence data equalise proportions, these taxa disproportionately contribute to the index, consistent with the expectations of hypothesis (ii). High abundances of mud-dwelling and littoral taxa are considered atypical and/or bad for this ST, indicating a strong impact of flow modification and fine sediment entry. Transformation to presence/absence data systematically upweighted the importance of the low-abundance mud-dwelling and littoral taxa, leading to the systematically lower scores. We observed the same effect for ST21_N and 21_S (lake outlets), for which only three core metrics are used to calculate the GDM (EPT [%], lake outlet typology index [equivalent to the GFI] and proportion of phytal taxa). However, the sample sizes were very low for these STs. Although only the proportion of phytal taxa (macrophyte-associated taxa) uses raw abundance data, this indicator has few core metrics (S3 Table), and account for 25% of the overall assessment result, which equals the impact of both abundance-based metrics combined in ST16. Hypothesis (ii) is further supported by the results for ST18 (small loess and loam-dominated lowland rivers), where 19% of the cases had a lower ESC, which is about half as many as in the other types analysed. For this ST, the proportion of

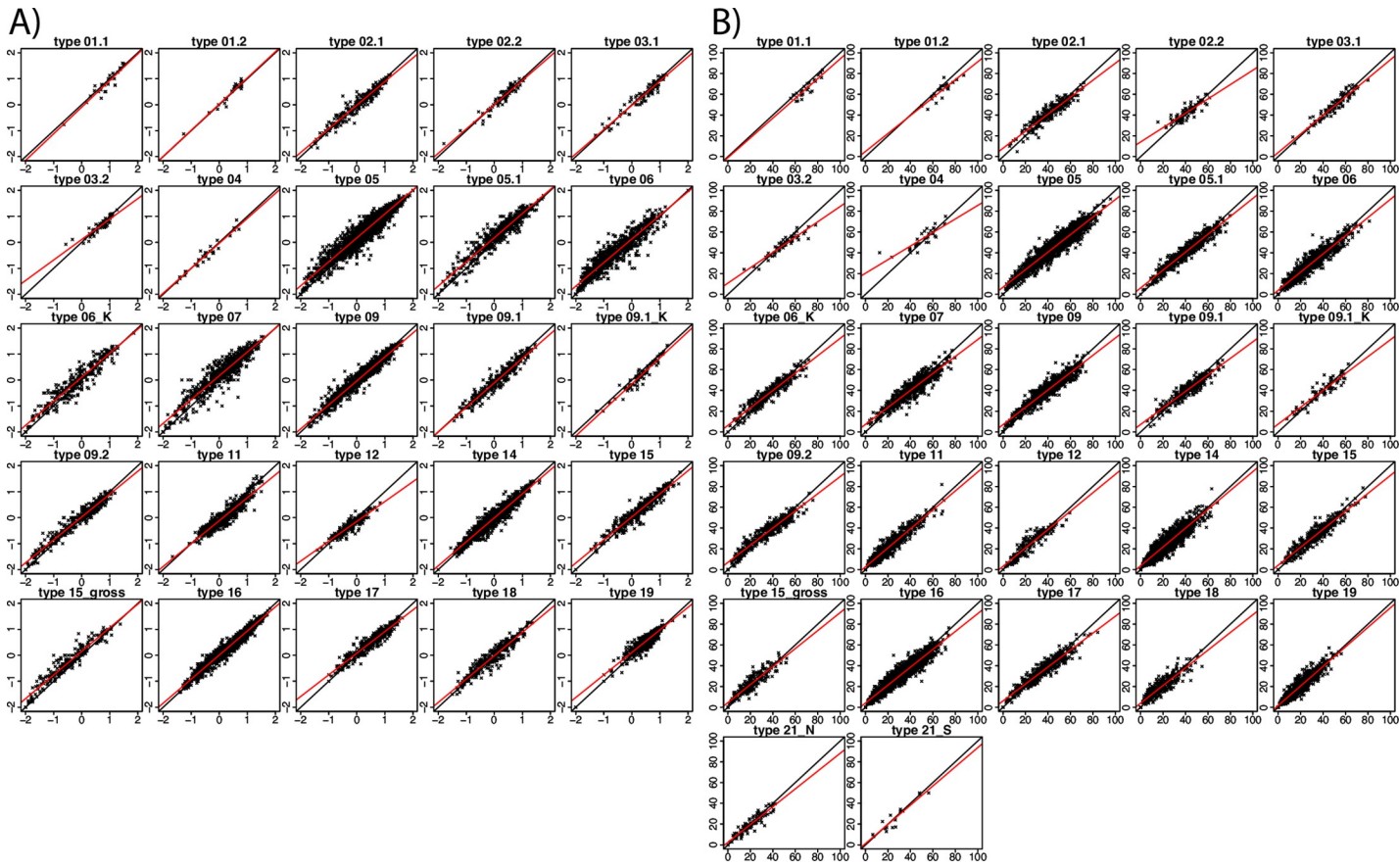

**Fig 5.** Regression analysis of presence/absence (*y*-axis) and abundance-based data (*x*-axis) for two metrics of the general degradation module: A) German fauna index, B) EPT[%].

littoral taxa is also used to calculate the GDM, but here four core metrics are used in the calculation and therefore the impact is only 12.5% (with 50% always owed to the GFI), considerably alleviating the effect. The high deviation in ST1.1 (23%) might be related to the low number of data points (n = 26).

We also found a notable bias for ST6 and 7 (small fine and coarse substrate-dominated calcareous highland rivers), where presence/absence data performed systematically better than abundance data (20% and 25%, respectively). Unlike ST16, a high proportion of epirhithral taxa is typical for these STs, thus pushing the values towards more positive classes. Typically, presence/absence data overestimate this proportion, leading to the results being systematically higher for the metric and thus the GDM. The effect might be weaker for the same reason as for ST18 because it only contributes 12.5% to the calculation of the GDM.

For ST10 and 20, many samples were evaluated more positively with presence/absence data than with abundance data (34% and 29%, respectively). Since there were almost no deviations when looking at the OPM, it is again obvious that the data type for this module has the greatest influence on ESC results. Since the only core metric used in these STs is the Potamon type index, for which results were highly correlated ($\rho$ = 0.94 and 0.95), the notable discrepancy has to be a class (border) assignment issue because correlations for the GDM were significantly lower ($\rho$ = 0.84 and 0.87) than those for the underlying metric.

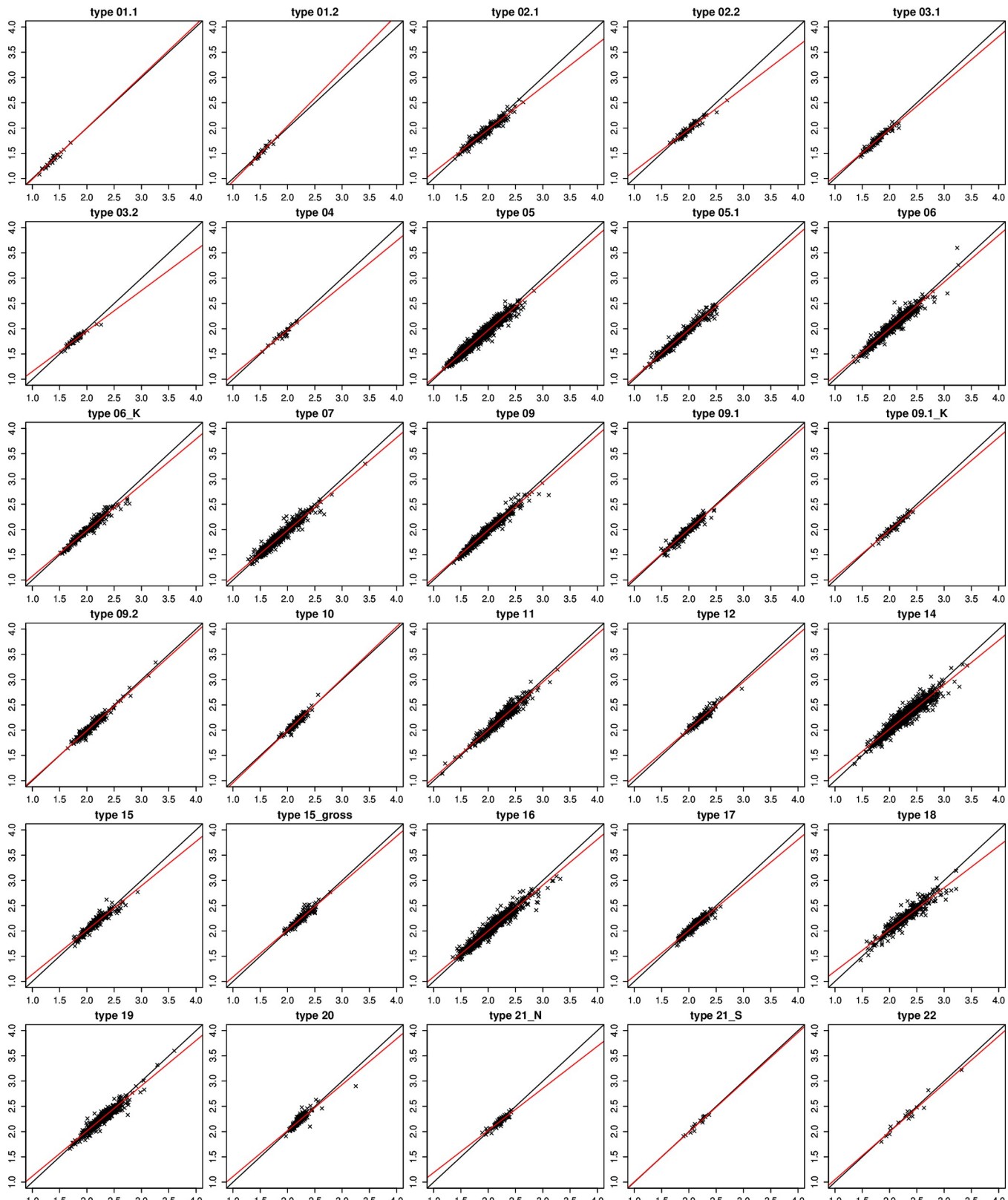

**Fig 6. Regression analysis of presence/absence (*y*-axis) and abundance-based data (*x*-axis) for the German saprobic index (organic pollution module, OPM).**

For ST5.1 (small, coarse substrate-dominated siliceous highland rivers), we obtained high correlations for all core metrics and even the GDM (ρ values of 0.95, 0.95, 0.96 and 0.97) while the overall results for the GDM show a high proportion of deviating values (20%). Here, again, we suspect that this is due to a class (border) assignment issue. In agreement with our initial hypothesis (ii), in most cases where presence/absence-based results differ from abundance-based results, one or more metrics directly relied on the abundance data used to calculate the GDM, leading to a positive correlation (see S3 Table). The class border issue, however, is independent of this.

## Limitations and prospects of DNA metabarcoding for aquatic bioassessment

Molecular methods can generate highly resolved taxa lists in a standardised way and in a short time [25]. Furthermore, they unlock a wealth of otherwise inaccessible information, such as information on hidden diversity [26,27], and can be used for the early detection of rare and protected or alien invasive species [28,29]. However, these types of data come with additional sources of variation that need to be considered [14]. For example, metabarcoding approaches can identify ingested prey items, thereby generating much larger and diverse taxa lists. These highly resolved taxonomic data are not necessarily applicable to standardised assessment methods, such as those of the WFD; for instance, chironomid species, which can easily be identified using DNA-based data, are not considered in the assessment systems of most countries [30]. Yet, from this high-resolution data all possible data sets required by any currently used WFD assessment method can be derived, for instance to comply with operational taxa lists at genus-level.

Most importantly, DNA metabarcoding can only be used to roughly infer abundance data. Abundance is a key parameter in ecology and is of great value for biomonitoring. Without abundance or biomass data, population trends cannot be picked up. Therefore, abundance as well as other quantitative data such as biomass, size etc. *per se* have immense value for understanding ecological dynamics and for establishing management or conservation strategies and need to be considered also in the future for various purposes (e.g. [31,32].

Our results, as well as those by Beentjes *et al*. [18] for Dutch lentic and lotic freshwaters and Wright-Stow and Winterbourne [17] for New Zealand streams, indicate that the inability to generate abundance data using DNA-based methods is of limited concern for ecological status inferences, as many metrics are based on incidence data. For metrics based on raw abundance data or on abundance classes, frequent minor deviations and occasional major deviations can be expected and this study showed that especially for those using raw abundance data (S3 Table) this is a concern. In the case of Dutch freshwaters, the calculated Ecological Quality Ratios (EQRs) use abundance information only to a minor extent [18]. Also, this information is provided in form of abundance classes, never as raw abundance information. This explains the generally higher agreement observed for the comparisons. While in the case of German streams the variation is at least in several cases substantially greater, available information can be used to train machine learning algorithms in order to calibrate both data types. The power of supervised machine learning for biomonitoring purposes and data intercalibration has been shown several times [33–36], yet, it is subject for future analyses to test the power on such data sets.

It should be noted that there are a few other incompatibilities between traditional and DNA-based methods. Some taxa are not detected due to primer bias [7,8] and gaps in reference databases. For the German operational taxa list, however, almost 90% of species-level records have barcodes. For most other countries in Europe, the thresholds for operational taxa

lists are quite high (>60% on average; [37]). A study of diatoms by DNA metabarcoding and traditional morphological approaches has shown that even with only 10% of taxa available, rather robust ESC results are possible [12], but country-specific deviations have also been reported [38]. An additional layer of variation may be due to differences in sample size. For WFD-conform macrozoobenthic-based assessment in Germany, typically a standardised number of specimens is picked and determined from the sample, which is either 350 or 700 specimens [20]. This is different in other countries and DNA metabarcoding of entire samples without prior sorting may thus lead to stronger deviations than the one described our *in silico* comparison here.

Therefore, we propose to test our approach in a more systematic fashion by simultaneously sampling, processing and identifying taxa with both the traditional method and the DNA-based method to validate the molecular method across STs or water bodies, in general. By this approach, gaps in reference libraries can also be filled. This will also demonstrate the capacities of different sampling strategies as well as different laboratory and bioinformatic approaches to process standard samples. Likewise, optimisation of the molecular workflow will reduce waiting time and overall costs, and thus make molecular approaches more attractive to policymakers and aquatic ecosystem managers [25,39]. In a next step, we will analyse how well ESC estimation based on presence/absence data performs across different national WFD-compliant tools and determine how the limitations of different molecular tools affect ESC/EQR inference. Thus, we aim to gauge whether our approach can be universally applied across Europe, as suggested by the results of this study. We are aware that the implementation of molecular tools will come at a cost and requires the utmost scientific rigour, but this is crucial for generating new data and comparing these with available data. So far, costs are in the same order of magnitude as for as traditional lab and identification procedures [7]. And if abundance estimates are continued to be obtained via traditional inspection or alternatively automated image recognition [40], no substantial cut in costs can be expected. However, the central incentive for including also genetic data should be the fundamentally improved resolution down to species or even population level [41] that can be obtained in a standardised fashion. While we explicitly encourage the development of new metrics and indices that make use of the full potential inherent in metabarcoding data [36], we emphasise the importance of properly evaluating the potential to link metabarcoding data to established indices and relate them to existing data. This should ideally be done in parallel with ESC inference based on current methods to develop well-founded and properly validated approaches that can be accepted both scientifically and from a water management perspective.

## Conclusions

Our results indicate that stream ESCs in Germany inferred from presence/absence data are, to a high degree, congruent with ESCs inferred from abundance data. However, the direct transformation of abundance-based ESC assessment into a presence/absence-based approach is not possible and will require a calibration e.g. by using supervised machine learning, based on direct comparisons of metabarcoding and traditional morpho-taxonomical data.

## Supporting information

**S1 Fig. Expected relationships between abundance (x-axis) and presence/absence-based (y-axis) metrics.**
(PDF)

**S2 Fig. Abundance metric analysis.** Deviation of the observed classification in the ecological status class (y-axis) in relation to the contribution of abundance information for the calculation of the module ($C_{AB}$).
(PDF)

**S3 Fig. Status class comparisons.** Comparison of abundance ('abd', black line with green confidence interval) and presence/absence (red dots) assessment results for 627 stream sites for which the transformation led to a status class shift from 2 to 3. Background shading indicates ecological status class intervals. These are geometrically defined for the %EPT and German fauna index (GFI) and therefore much narrower than for the German saprobic index.
(PDF)

**S1 Table. Overview of metrics used for assessment of the 30 different stream types.** Metrics that do not use abundance data are shown in green, metrics that use abundance classes are shown in yellow, and metrics that use raw abundance data are shown in red. HMWB = Heavily modified water bodies. Abbreviations are listed. For further reference see http://www.fliessge waesserbewertung.de/kurzdarstellungen/bewertung/
(XLSX)

**S2 Table. Correlation analysis.** Overview of correlation values (Spearman's ρ) for the ecological status class (ESC) and its three (two) underlying assessment modules, calculated with abundance data and presence/absence data.
(XLSX)

**S3 Table. Correlation between different metrics using abundance and presence/absence data (Spearman's ρ).** Metrics that do not use abundance data are shown in green, metrics that use abundance classes are shown in yellow, and metrics that use raw abundance data are shown in red. Values are only shown in cells for stream types for which the metric is actually used.
(XLSX)

**S4 Table. Status class shifts.** Shifts from good to moderate (2→3) ecological status class, or vice versa (3 → 2), for the 30 analysed stream types after transformation to presence/absence data.
(XLSX)

**S1 File. Programming code used for data analysis.**
(ZIP)

## Acknowledgments

We thank the German states and LAWA (Bund/Länder-Arbeitsgemeinschaft Wasser) and the state agencies for providing data for analyses and helpful discussion (Bavaria: Bavarian Environment Agency, Baden-Wuerttemberg: Baden-Württemberg State Institute fort he Environment, Survey and Nature Conservation, Hesse: Hessian Agency for Nature Conservation, Environment and Geology, Lower Saxony: Lower Saxony Water Management, Coastal Defence and Nature Conservation Agency, North Rhine-Westphalia: North Rhine-Westphalia State Agency for Nature, Environment and Consumer Protection, Saxony: Saxon State Office for the Environment, Agriculture and Geology, Saxony-Anhalt: State Agency for Flood Defence and Water Management of Saxony-Anhalt, Schleswig-Holstein: State Agency for Agriculture, Environment and Rural Areas Schleswig-Holstein). We furthermore thank Jens Arle (UBA Dessau) for specific comments on aspects of the manuscript. This work was

conducted within the framework of EU COST Action DNAqua-Net (CA15219) and German Barcode of Life 2 project (project 01LI1501K) to Florian Leese.

## Author Contributions

**Conceptualization:** Dominik Buchner, Arne J. Beermann, Simon Vitecek, Daniel Hering, Florian Leese.

**Data curation:** Dominik Buchner, Peter Rolauffs.

**Formal analysis:** Dominik Buchner, Alex Laini, Florian Leese.

**Investigation:** Dominik Buchner, Arne J. Beermann, Alex Laini, Peter Rolauffs, Simon Vitecek, Daniel Hering.

**Methodology:** Dominik Buchner, Alex Laini, Simon Vitecek.

**Project administration:** Florian Leese.

**Resources:** Florian Leese.

**Software:** Dominik Buchner.

**Supervision:** Florian Leese.

**Visualization:** Dominik Buchner.

**Writing – original draft:** Florian Leese.

**Writing – review & editing:** Dominik Buchner, Arne J. Beermann, Alex Laini, Peter Rolauffs, Simon Vitecek, Daniel Hering.

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
