## [Decision Letter · Decision Letter 0]

18 Sep 2019

PONE-D-19-16057

Analysis of 13,000 benthic invertebrate samples from German streams reveals minor deviations in ecological status class between abundance and presence/absence data

PLOS ONE

Dear Prof Dr Leese,

Thank you for submitting your manuscript to PLOS ONE. After careful consideration, we feel that it has merit but does not fully meet PLOS ONE’s publication criteria as it currently stands. Therefore, we invite you to submit a revised version of the manuscript that addresses the points raised during the review process.

I have now received the comments of three external reviewers and as you can see they have very contrasting viewpoints and raised different concerns, particularly reviewer 2.

Reviewers 1 and 3 are very positive and have only minor to moderate suggestions. Reviewer 1 suggests to clarify some analyses and to present the data, at least by including a supplementary figure. On the other hand, reviewer 2 is rather skeptical of the merit of the manuscript as a direct transformation is very challenging and potentially biased without a correction factor.

We would appreciate receiving your revised manuscript by Nov 02 2019 11:59PM. To enhance the reproducibility of your results, we recommend that if applicable you deposit your laboratory protocols in protocols.io, where a protocol can be assigned its own identifier (DOI) such that it can be cited independently in the future. For instructions see: http://journals.plos.org/plosone/s/submission-guidelines#loc-laboratory-protocols

We look forward to receiving your revised manuscript.

Kind regards,

Fabrizio Frontalini

Academic Editor

PLOS ONE

Journal Requirements:

Additional Editor Comments (if provided):

Reviewers' comments:

Reviewer's Responses to Questions

**Comments to the Author**

1. Is the manuscript technically sound, and do the data support the conclusions?

Reviewer #1: Yes

Reviewer #2: Partly

Reviewer #3: Yes

2. Has the statistical analysis been performed appropriately and rigorously? 

Reviewer #1: Yes

Reviewer #2: Yes

Reviewer #3: Yes

3. Have the authors made all data underlying the findings in their manuscript fully available?

Reviewer #1: Yes

Reviewer #2: No

Reviewer #3: No

4. Is the manuscript presented in an intelligible fashion and written in standard English?

Reviewer #1: Yes

Reviewer #2: Yes

Reviewer #3: Yes

5. Review Comments to the Author

Reviewer #1: The authors present a well-written and interesting paper on the effects of transforming abundance data into presence/absence data for freshwater quality monitoring. Their dataset is impressive, and the approach of using different metrics, which deal with different kinds of abundance data, makes that it’s a valuable contribution to the discussions surrounding the adaptation of quality metrics to better incorporate molecular data. My only real concern with the paper is that the relevance of two of the analyses performed is not entirely clear, mostly due to lack of data or inclusion in the general discussion.

First, the authors describe a method to evaluate the impact of the transformation, by calculation the percentage of abundance-reliant data for each of the stream types. This in itself is interesting, but the results are only summarized in a single sentence (lines 269-271) and no data is shown. It would be good to include the data or a visualization of the data in a supplementary figure at least, maybe even in the main article. Second, the detailed analysis of ST5 and ST14 to see if deviations are different for the highest and lowest ESCs has merit, but the results are only presented in the results section and not mentioned any further in the discussion of the paper. As a reader, I fail to see if it is either logical or of any impact whether some metrics (I think most if not all of them are based on abundance classes) have significantly higher of lower deviations when data is transformed. I would urge the authors to discuss the relevance of these results in the discussion section. As of now this detailed analysis seems to be an afterthought that is not fully incorporated into the paper.

Besides these two main points, I have a number of smaller comments listed below. All in all, I think the authors did a good job of presenting a nation-wide study into the effects of transforming surveys into presence/absence data, and it will surely be a valuable contribution and inspiration for other nations to similarly assess the viability of adopting current assessment methodology to make use of metabarcoding data.

Line 39: There is a stray “4.”, likely left over from when the abstract was a numbered list.

Line 77: Consider the use of a comma after “water bodies” or restructuring the sentence. It now reads as if different pressures also affect biogeography, which I assume they don’t.

Line 107-111: Two papers mentioned here in passing (references 18 and 19) can be considered systematic and large-scale in my opinion. They both directly address the question whether data transformation into presence/absence is feasible, and use a large number of data points (roughly 1800 and 700, respectively). Can the authors maybe come back to these papers in the discussion, and compare results? As it is written now, it feels as if the authors try to increase the novelty of their own work by stating these papers merely “suggest” a general coherence.

Line 145: How many abundance classes are used in traditional metrics, and how are they classified? Including this information in the paper would allow readers (and authors) to evaluate whether observed deviations are to be expected or not. Deviations would be different if 3 classes were used, or 12 classes.

Line 235-236: What is meant by falling “outside the threshold criterion”?

Line 237 / Table 1: Consider putting these tables in the same order as Figures 2-4.

Line 237 / Table 1: The numbers in these tables do not add up. ESC has a total of 13312 samples (which matches the materials and methods), but OPM has a total of 13429, and GDM has a total of 13293.

Line 237 / Table 1: It should be noted that for the original assessments using abundance data ~77% of OPM are either “high” or “good”, whereas only ~30% of GDM scores in those two categories. This should be included in the discussion of the paper, especially since the change in OPM seems limited by the data transformation. It means that changes in GDM will in most cases be reflected directly in changes in ESC (as observed in figures 2 and 3), due to the facts that (1) for the ESC the lowest of the two sub-scores (OPM/GDM) counts as the final score, and (2) GDM is likely the lowest in most cases (perhaps the authors can calculate the percentage of cases?). Lower GDM almost would then always lead to lower ESC (unless the OPM was lower to start with, but that seems unlikely), and higher GDM would usually lead to higher ESC, when OPM was higher than original GDM.

Line 252-255: 28 stream types are mentioned, but figure 5A only has 25 panels, and 5B only has 27. Are some STs left out of the analysis, and if so, why? I’m also assuming that 5A is GFI and 5B is EPT[%], but this could be clarified in the text and especially in the legend of the figure.

Line 255-258: Here the authors also mention OPM results in relation to figure 5. Is any part of the figure representing OPM results?

Line 261-262: Metrics using p/a data will not change when transforming abundance to p/a data. This is rather logical, and would not require stating, let alone using statistics to calculate a “perfect” correlation of 1.

Line 269-271: This is the only mention of any results for the contribution of abundance data. Please show the data, or a figure representing the data.

Line 274: It is unclear what the authors mean by “systematic errors”. I’m assuming it means shifts in ESC, from the paragraph that follows, but this could be made clearer.

Line 274-280: Authors state a “limited number of cases”, but then cite shifts that amount to roughly 14% of the data points. In total one fourth of all ESCs changed (line 308). I feel this is more than just a “limited number”. Consider rephrasing to better represent the amount of class shifts.

Line 290-292: This sentence is hard to read. It would be better if it were reversed, e.g. “we observed significantly lower deviations in the metrics EPT [%] and % hyporhithral, and significantly higher deviations for SI, rheo index and GFI for ‘high’ ESCs”.

Line 327 and throughout the manuscript: Spearman’s correlation values are denoted with ρ (rho) or rs.

Line 237-333: At first, the pelagic and littoral taxa have low abundances, but later they have high abundances? The authors probably refer to the transformation changing the proportions of pelagic/littoral versus non-pelagic/littoral taxa, so it might be better to talk about “proportions” rather than “abundances”.

Line 337: Shouldn’t this be 16.7% instead of 33%? As GFI is already 50% of the GDM.

Line 340: This is 19% in figure 2.

Line 343: “significantly”. Has this been statistically tested? If not, consider using a different word here.

Line 346: What do authors mean by “the end of the spectrum”?

Line 348: These percentages also do not match figure 2 (nor figure 3).

Line 359: The class assignment is mentioned here. Have the authors also looked into raw scores, or just the final class assignments? It is likely that many of the samples that changed class were already close to a class border. In case the authors have looked at this, it might be worth including in the discussion.

Line 364-367: It is unclear what the authors mean with this sentence, and which correlation they refer to.

Line 377-380: On the contrary, higher resolution data is compatible, since it can always be translated into lower resolution data. E.g., all chironomid species could just be tallied under “Chironomidae” if the standard method only requires family-level information.

Line 381: Consider removing/replacing the word “Importantly”, since the previous sentence already starts with “Most importantly”.

Line 404: I’m not sure that more DNA extraction comparisons are all that’s needed. There are many other factors that play a role in metabarcoding, such as primers and pre-processing of material that may have much greater impact on the resulting taxa lists.

Line 405-407: Consider including a reference to e.g. Aylagas et al 2018, in which the cost and time reduction was calculated.

Line 554 / Figure 3: The number of samples is not the same in figure and legend.

Line 568-569 / Figure 5: Please state what figure A and B are representing. Maybe it would be good to include the slope/correlation values in the panels of the figures, instead of presenting them in a separate supplemental file.

Line 584 / S1 Table: Please don’t use abbreviations in the metric names, or provide abbreviations in the description instead of referring to a German website. Also, the names of the metrics in table S1 don’t match the names of the metrics in table S3.

Line: 587-588 / S2 Table: Description is a little short. Correlation values of what? Are these correlations between abundance and p/a data? If so, would it be possible to combine tables S2 and S3 into one supplement? Both tables show correlation values for GDM, but they don’t match?

Reviewer #2: Review remarks „Analysis of 13,000 benthic invertebrate samples from German streams reveals minor deviations in ecological status class between abundance and presence/absence data” submitted as PONE-D-19-16057to PLOS ONE by Buchner et. al. 2019

General comments:

1. The study presents the results of original research.

Yes.

2. Results reported have not been published elsewhere.

Yes.

3. Experiments, statistics, and other analyses are performed to a high technical standard and are described in sufficient detail.

Partly (see specific comments).

4. Conclusions are presented in an appropriate fashion and are supported by the data.

Partly (see specific comments).

5. The article is presented in an intelligible fashion and is written in standard English. Yes

6. The research meets all applicable standards for the ethics of experimentation and research integrity.

Yes.

7. The article adheres to appropriate reporting guidelines and community standards for data availability.

Yes.

Specific comments:

Line 42 -46 Statement “Systematic stream type-specific deviations were found and 43 differences between abundance and presence/absence data were most prominent for stream 44 types where abundance information contributed directly to one or several metrics of the 45 general degradation module.” not underpinned by results.

Line 47- 48 “The systematic decrease in scores was observed, even when considering simulated confidence intervals for abundance data.” not underpinned by results.

Line 129 -121 “,…thus representative of a variety of approaches developed for other BQEs or in other countries” not underpinned by results.

Line 153 “transformed to presence/absence data”: not suffiently described, how was the transformation done?

Line 197 – 208: Simulation of taxon lists: the role of this exercise is not clear, because the results are neither presented nor discussed.

Line 261 – 262: Metrics that use presence/absence data only were perfectly correlated when calculated with presence/absence data instead of abundance data (Spearman’s Rho = 1, Table S3). Looks like a circular reasoning.

Line 281 – 286: Detailed analyses based on simulated abundance data for the 627 cases etc…. It is unclear, how was this done?

Line 288 – 299 Why a detailed analysis of ST5 and ST14 only? What was the rationale in relation to the hypothesis?

Line 324 – 327 While strong positive correlations between data types were found for the GFI, the number of Trichoptera species and EPT [%], the results for proportions of pelagic and littoral taxa were poorly correlated (r = 0.41 and 0.56). What are pelagic and littoral invertebrates in streams?

Line 390 - 391: However, correction factors can be included in the assessment formulas or class boundaries. Is this really meaningful and contradictory to arguments in Lines 98 through 102?

Lines 420 -426: This arguments are repeated in the conclusions.

Lines 422 – 424: However, the direct transformation of abundance-based ESC assessment into a presence/absence-based approach is not possible and will require correction factors determined from comparisons of abundance based and presence/absence data.

Because of theoretical and mathematical reasons, there can never be a “direct transformation” of both approaches. A convincing evidence from the analysis and argument is missing, why large efforts should now be implemented to determine “correction factors”, while the authors at the same time advocate to replace the classical taxa × abundance matrices with taxa × presence/absence matrices obtained by the molecular characterisation of communities and entirely new sets of metrics and indices based on molecular data.

Reviewer #3: Review: Analysis of 13,000 benthic invertebrate samples from German streams reveals minor deviations in ecological status class between abundance and presence/absence data

The Manuscript(MS) deals with a very interesting topic, whether presence/absence data applicable for present monitoring activity and if it gives the same results as in case of abundance based metric.

TH MS is clear, contains a lot of efforts and analysis to support it findings. The used statistical tools are adequate to answer the given scientific questions.

I have just minor comments on the MS

Title: Why not put the exact number 13,312 in the title?

L31 “cheap and accurate” this phrase only occurs in the abstract not mentioned in the Introduction please give a reference to why eDNA is cheaper and accurate than traditional methods.

L 39 “4.” not needed

L 71 “The resulting score ..” – change to The resulting score the Ecological Quality Ratio (EQR) is then. …

Generally, the term EQR is used for assessment more often.

L77 If there are intercalibrated results compared among multiple countries why not focus only on these types and these metrics which were the intercalibration? The MS is only focusing on German stream types but with the use of a common intercalibration metric, the results could achieve more broad interest.

L 150 “The total data set contained 13,401 samples obtained from monitoring sites in 1985–2013, „

The question is why to use all of this data? This assumes that the data is the same quality from 1985 to 2013, but the WFD is much younger maybe the quality could differ from years to years and the differences of presence/absence vs. abundance assessment could be interfered.

Does it also occur that in some types there is an order of magnitude of the analysed samples than in others why not standardize or limit the number of observation among types?

Why not limit the analyses just for a smaller time-frame, like the second Water Basin Management plan frame.

L 154 What was the cause of errors in the calculation? Is it just mathematical or systematic for a given typology?

L 159 Appendix S1 was not included in the MS

L 255 FIG 5 should be in the supplement, not in the MS. It is not informative in this form nor the min and max values could be resolved.

L 279 (Table S4) This is one of the most informative tables, it should be in the MS text not in the supplement (include the number of unchanged “good” sites not only the shifts)

Fig 1. branch of “excluding data-… “ not needed, use the 13,312 samples

Table S4 include the number of unchanged “good” sites not only the shifts

6. PLOS authors have the option to publish the peer review history of their article (what does this mean?). If published, this will include your full peer review and any attached files.

Reviewer #1: Yes: Kevin Beentjes

Reviewer #2: No

Reviewer #3: Yes: Gabor Varbiro

---

## [Author Response · Author response to Decision Letter 0]

15 Nov 2019

Dear editor, dear reviewers,

Thanks for the detailed and constructive comments on our manuscript. We have carefully revised our manuscript along the lines of your suggestions. Below, we provide a detailed point-by-point response to every aspect raised. Reviewer remarks that we revised as suggested are highlighted in green. Remarks where we did not follow the reviewers’ suggestions are highlighted in yellow. Here, we gave detailed explanations on why we did not incorporate them. Please be aware that the colour marking is only available in the uploaded document - here the responses are without colour code.

Best regards

Florian Leese, Dominik Buchner and all co-authors

-------

Reviewer #1: The authors present a well-written and interesting paper on the effects of transforming abundance data into presence/absence data for freshwater quality monitoring. Their dataset is impressive, and the approach of using different metrics, which deal with different kinds of abundance data, makes that it’s a valuable contribution to the discussions surrounding the adaptation of quality metrics to better incorporate molecular data. My only real concern with the paper is that the relevance of two of the analyses performed is not entirely clear, mostly due to lack of data or inclusion in the general discussion.

First, the authors describe a method to evaluate the impact of the transformation, by calculation the percentage of abundance-reliant data for each of the stream types. This in itself is interesting, but the results are only summarized in a single sentence (lines 269-271) and no data is shown. It would be good to include the data or a visualization of the data in a supplementary figure at least, maybe even in the main article. 

Reply: Thanks for the constructive remarks. We added the requested changes by creating a new figure (Figure S2). This is now mentioned in the respective section of the results and briefly discussed. 

Second, the detailed analysis of ST5 and ST14 to see if deviations are different for the highest and lowest ESCs has merit, but the results are only presented in the results section and not mentioned any further in the discussion of the paper. As a reader, I fail to see if it is either logical or of any impact whether some metrics (I think most if not all of them are based on abundance classes) have significantly higher of lower deviations when data is transformed. I would urge the authors to discuss the relevance of these results in the discussion section. As of now this detailed analysis seems to be an afterthought that is not fully incorporated into the paper.

Reply: We decided to remove the analysis. While we find several significant differences (e.g. Stream Type 5, Deviation for Saprobic Index in Class 1: 0.0259 vs. Deviation in Class 2-5: 0.0361; p=0.0002), all these are extremely small in terms of absolute numbers and also differ among stream types. We thus think that the analysis adds no clear value to the comparison done in this MS it was – as stated – more an ‘afterthought’.

Besides these two main points, I have a number of smaller comments listed below. All in all, I think the authors did a good job of presenting a nation-wide study into the effects of transforming surveys into presence/absence data, and it will surely be a valuable contribution and inspiration for other nations to similarly assess the viability of adopting current assessment methodology to make use of metabarcoding data.

Reply: Thanks for the positive remarks.

Line 39: There is a stray “4.”, likely left over from when the abstract was a numbered list.

Reply: done

Line 77: Consider the use of a comma after “water bodies” or restructuring the sentence. It now reads as if different pressures also affect biogeography, which I assume they don’t.

Reply: Thanks. We rephrased as follows: “These differences are rooted in local monitoring traditions, different biogeographies and different pressures affecting the water bodies.”

Line 107-111: Two papers mentioned here in passing (references 18 and 19) can be considered systematic and large-scale in my opinion. They both directly address the question whether data transformation into presence/absence is feasible, and use a large number of data points (roughly 1800 and 700, respectively). Can the authors maybe come back to these papers in the discussion, and compare results? As it is written now, it feels as if the authors try to increase the novelty of their own work by stating these papers merely “suggest” a general coherence.

Reply: We now discuss this in a new paragraph in the discussion.

Line 145: How many abundance classes are used in traditional metrics, and how are they classified? Including this information in the paper would allow readers (and authors) to evaluate whether observed deviations are to be expected or not. Deviations would be different if 3 classes were used, or 12 classes.

Reply: We now state this in this sentence as well as in Table S1.

Line 235-236: What is meant by falling “outside the threshold criterion”?

Reply: We added the requested information: “(i.e. at least ESC = 2)”

Line 237 / Table 1: Consider putting these tables in the same order as Figures 2-4.

Reply: Well spotted, thanks. We change the order of the figures to make it consistent.

Line 237 / Table 1: The numbers in these tables do not add up. ESC has a total of 13312 samples (which matches the materials and methods), but OPM has a total of 13429, and GDM has a total of 13293.

Reply: Well-spotted. We corrected the numbers. Please note, that the GDM only has 13295 associated data points since it is not calculated for the 17 streams of type 22. 

Line 237 / Table 1: It should be noted that for the original assessments using abundance data ~77% of OPM are either “high” or “good”, whereas only ~30% of GDM scores in those two categories. This should be included in the discussion of the paper, especially since the change in OPM seems limited by the data transformation. It means that changes in GDM will in most cases be reflected directly in changes in ESC (as observed in figures 2 and 3), due to the facts that (1) for the ESC the lowest of the two sub-scores (OPM/GDM) counts as the final score, and (2) GDM is likely the lowest in most cases (perhaps the authors can calculate the percentage of cases?). Lower GDM almost would then always lead to lower ESC (unless the OPM was lower to start with, but that seems unlikely), and higher GDM would usually lead to higher ESC, when OPM was higher than original GDM.

Reply: We checked the data, added the newly calculated results (its 95% for untransformed and transformed data) to both results (paragraph on GDM) and to the first paragraph of discussion.

Line 252-255: 28 stream types are mentioned, but figure 5A only has 25 panels, and 5B only has 27. Are some STs left out of the analysis, and if so, why? I’m also assuming that 5A is GFI and 5B is EPT[%], but this could be clarified in the text and especially in the legend of the figure.

Reply: The different metrics / modules are not calculated for all stream types (see Tab. S1). Therefore, number of stream types shown differs.

Line 255-258: Here the authors also mention OPM results in relation to figure 5. Is any part of the figure representing OPM results?

Reply: The plots were missing in figure 5 and are now added as figure 6.

Line 261-262: Metrics using p/a data will not change when transforming abundance to p/a data. This is rather logical, and would not require stating, let alone using statistics to calculate a “perfect” correlation of 1.

Reply: Thanks, changed. Now reads: “As expected, metrics that use presence/absence of taxa (e.g. number of Trichoptera taxa) were perfectly correlated when calculated with presence/absence data (Spearman’s ρ = 1, Table S3).” We want to keep this part in since it shows that there are 3 categories of metrics, of which only two are problematic when calculating metrics and modules with pa data.

Line 269-271: This is the only mention of any results for the contribution of abundance data. Please show the data, or a figure representing the data.

Reply: Thanks, as mentioned above in response to the main critique we have now added a new figure (Fig. S2) and discuss this also briefly.

Line 274: It is unclear what the authors mean by “systematic errors”. I’m assuming it means shifts in ESC, from the paragraph that follows, but this could be made clearer.

Reply: Now reads “differ” instead of systematic errors. Since we observe differences in both directions, they can’t be considered systematic.

Line 274-280: Authors state a “limited number of cases”, but then cite shifts that amount to roughly 14% of the data points. In total one fourth of all ESCs changed (line 308). I feel this is more than just a “limited number”. Consider rephrasing to better represent the amount of class shifts.

Reply: We agree and rephrase the sentence, which now reads: “We found that presence/absence-based ESC estimates differed in 23% of the observed cases. Of major practical importance are shifts from ESC 2 (‘good’) to ESC 3 (‘moderate’, i.e. not meeting the requirements of the WFD). …”

Line 290-292: This sentence is hard to read. It would be better if it were reversed, e.g. “we observed significantly lower deviations in the metrics EPT [%] and % hyporhithral, and significantly higher deviations for SI, rheo index and GFI for ‘high’ ESCs”.

Reply: This sentence was removed (whole analysis).

Line 327 and throughout the manuscript: Spearman’s correlation values are denoted with ρ (rho) or rs.

Reply: done

Line 237-333: At first, the pelagic and littoral taxa have low abundances, but later they have high abundances? The authors probably refer to the transformation changing the proportions of pelagic/littoral versus non-pelagic/littoral taxa, so it might be better to talk about “proportions” rather than “abundances”.

Reply: We agree that the paragraph starting L321 in the old, L318 in the new version, was misleading and rephrased it. It now reads “This can be explained by the large number of mud-dwelling and littoral taxa, albeit with low abundances, and the fact that both metrics use raw abundance data (Table S3, red shading). As presence/absence data equalise proportions, these taxa disproportionately contribute to the index, consistent with the expectations of hypothesis (ii). High abundances of mud-dwelling and littoral taxa are considered atypical and/or bad for this ST, indicating a strong impact of flow modification and fine sediment entry. Transformation to presence/absence data systematically upweighted the importance of the low-abundance mud-dwelling and littoral taxa, leading to the systematically lower scores.”

Line 337: Shouldn’t this be 16.7% instead of 33%? As GFI is already 50% of the GDM.

Reply: In fact both numbers were wrong. Since the GFI (or the equivalent in the case of ST21) takes up 50% the remaining 50% are divided between the two remaining core metrics leading to value of 25%. This is now corrected.

Line 340: This is 19% in figure 2.

Reply: Thanks for mentioning this, it is correct now.

Line 343: “significantly”. Has this been statistically tested? If not, consider using a different word here.

Reply: Now reads “considerably”.

Line 346: What do authors mean by “the end of the spectrum”?

Reply: The term unnecessarily complicates the facts and we removed this. The section now reads: “We also found a notable bias for ST6 and 7 (small fine and coarse substrate-dominated calcareous highland rivers), where presence/absence data performed systematically better than abundance data (20% and 25%, respectively).”

Line 348: These percentages also do not match figure 2 (nor figure 3).

Reply: Thanks again, changed to 20 and 25%.

Line 359: The class assignment is mentioned here. Have the authors also looked into raw scores, or just the final class assignments? It is likely that many of the samples that changed class were already close to a class border. In case the authors have looked at this, it might be worth including in the discussion.

Reply: Correlations for raw scores are shown in Table S3. Since they are highly correlated in most cases (always higher than the underlying module) a class border issue is highly probable. When looking at the results of the simulation exercise it can be seen that the class border often interferes with the confidence intervals for the metrics. 

Line 364-367: It is unclear what the authors mean with this sentence, and which correlation they refer to.

Reply: We added the reference to Table S3 here.

Line 377-380: On the contrary, higher resolution data is compatible, since it can always be translated into lower resolution data. E.g., all chironomid species could just be tallied under “Chironomidae” if the standard method only requires family-level information.

Reply: We added a sentence to highlight this: “Yet, from this high resolution data all possible data sets required by any currently used WFD assessment method can be derived, for instance to comply with operational taxa lists at genus-level.”

Line 381: Consider removing/replacing the word “Importantly”, since the previous sentence already starts with “Most importantly”.

Reply: Deleted.

Line 404: I’m not sure that more DNA extraction comparisons are all that’s needed. There are many other factors that play a role in metabarcoding, such as primers and pre-processing of material that may have much greater impact on the resulting taxa lists.

Reply: Indeed, this level of detail is also not needed here. We removed the sentence.

Line 405-407: Consider including a reference to e.g. Aylagas et al 2018, in which the cost and time reduction was calculated.

Reply: Done

Line 554 / Figure 3: The number of samples is not the same in figure and legend.

Reply: Done.

Line 568-569 / Figure 5: Please state what figure A and B are representing. Maybe it would be good to include the slope/correlation values in the panels of the figures, instead of presenting them in a separate supplemental file.

Reply: A and B are now stated in the legend. We chose not to include the slope since it makes the plots less readable. The general tendency can be seen well without slopes. 

Line 584 / S1 Table: Please don’t use abbreviations in the metric names, or provide abbreviations in the description instead of referring to a German website. Also, the names of the metrics in table S1 don’t match the names of the metrics in table S3.

Reply: We added definitions to the abbreviations to Table S1.

Line: 587-588 / S2 Table: Description is a little short. Correlation values of what? Are these correlations between abundance and p/a data? If so, would it be possible to combine tables S2 and S3 into one supplement? Both tables show correlation values for GDM, but they don’t match?

Reply: Yes, these are correlations between abundance and p/a data. We extended the caption text as requested. Concerning combination. We consider it important to keep module (S2) and metric (S3) separate, as we also discuss them separately. Regarding the mismatch: The module shows corrections when taking the five classes as the basis (S2). For the metrics, the correlation is calculated for the GDM prior to assigning it to classes, i.e. the metric has a continuous value range from 0 to 1.

---

Reviewer #2: Review remarks „Analysis of 13,000 benthic invertebrate samples from German streams reveals minor deviations in ecological status class between abundance and presence/absence data” submitted as PONE-D-19-16057to PLOS ONE by Buchner et. al. 2019

General comments:

1. The study presents the results of original research.

Yes.

2. Results reported have not been published elsewhere.

Yes.

3. Experiments, statistics, and other analyses are performed to a high technical standard and are described in sufficient detail.

Partly (see specific comments).

4. Conclusions are presented in an appropriate fashion and are supported by the data.

Partly (see specific comments).

5. The article is presented in an intelligible fashion and is written in standard English. Yes

6. The research meets all applicable standards for the ethics of experimentation and research integrity.

Yes.

7. The article adheres to appropriate reporting guidelines and community standards for data availability.

Yes.

Specific comments:

Line 42 -46 Statement “Systematic stream type-specific deviations were found and differences between abundance and presence/absence data were most prominent for stream types where abundance information contributed directly to one or several metrics of the general degradation module.” not underpinned by results.

Reply: It actually is underpinned by the results, but we agree that too little information was provided. To more explicitly show that the statement is underpinned by data we added Figure S2, added a new sentence to results / discussion (see comment above to reviewer 1). Results are now shown in Figure S2 where it can be clearly seen that the mismatch is bigger the more abundance data is used in the calculation of the underlying module. Also mentioned in results and discussion now.

Line 47- 48 “The systematic decrease in scores was observed, even when considering simulated confidence intervals for abundance data.” not underpinned by results.

Reply: To highlight this result we changed Figure S3. Systematic decrease is now indicated by a line fitted to the dots using a linear model. It can be seen that values calculated with pa-data score systematically lower. 

Line 129 -121 “,…thus representative of a variety of approaches developed for other BQEs or in other countries” not underpinned by results.

Reply: We added a citation to Birk et al. 2012.

Line 153 “transformed to presence/absence data”: not suffiently described, how was the transformation done?

Reply: Now reads: “Abundances in the raw data were replaced by either 1 (presence) or 0 (absence) for this transformation”

Line 197 – 208: Simulation of taxon lists: the role of this exercise is not clear, because the results are neither presented nor discussed.

Reply: The results are shown in Figure S3 and described from Line 280-285. 

Line 261 – 262: Metrics that use presence/absence data only were perfectly correlated when calculated with presence/absence data instead of abundance data (Spearman’s Rho = 1, Table S3). Looks like a circular reasoning.

Reply: Changed. It now reads: “As expected, metrics that use presence/absence of taxa (e.g. number of Trichoptera taxa) were perfectly correlated when calculated with presence/absence data (Spearman’s ρ = 1, Table S3). Metrics that use abundance classes, such as the GFI, saprobic index, or rheo index, showed generally strong and significant correlations (mean Spearman’s ρ = 0.93, range: 0.89–0.96) between both data types.”

Line 281 – 286: Detailed analyses based on simulated abundance data for the 627 cases etc…. It is unclear, how was this done?

Reply: Please take a look at line 191-203. Variation in abundance was estimated by using the abundance of taxa as a mean for a zero truncated Poisson distribution from which new datapoints were generated. 

Line 288 – 299 Why a detailed analysis of ST5 and ST14 only? What was the rationale in relation to the hypothesis?

Reply: Thanks for highlighting this point. The analysis was a by-product and plays no central role with respect to the analyses. We deleted all sections on this analysis.

Line 324 – 327 While strong positive correlations between data types were found for the GFI, the number of Trichoptera species and EPT [%], the results for proportions of pelagic and littoral taxa were poorly correlated (r = 0.41 and 0.56). What are pelagic and littoral invertebrates in streams?

Reply: We added the definitions to the MS (pelagic = mud-dwelling; littoral = inhabiting lentic shallow water habitats). Regarding the correlations we added Fig. S2 and a more detailed discussion of the results.

Line 390 - 391: However, correction factors can be included in the assessment formulas or class boundaries. Is this really meaningful and contradictory to arguments in Lines 98 through 102?

Reply: Good point. Removed the correction factors and suggest to use the data with a supervised machine learning approach. 

Lines 420 -426: This arguments are repeated in the conclusions.

Reply: Lines 420 – 426 are the conclusions. We are not sure what the reviewer means here. However, we rephrased the conclusions slightly to focus them.

Lines 422 – 424: However, the direct transformation of abundance-based ESC assessment into a presence/absence-based approach is not possible and will require correction factors determined from comparisons of abundance based and presence/absence data.

Because of theoretical and mathematical reasons, there can never be a “direct transformation” of both approaches. A convincing evidence from the analysis and argument is missing, why large efforts should now be implemented to determine “correction factors”, while the authors at the same time advocate to replace the classical taxa × abundance matrices with taxa × presence/absence matrices obtained by the molecular characterisation of communities and entirely new sets of metrics and indices based on molecular data.

Retry: Agree; we removed discussion about correction factors and suggest to use the data with a supervised machine learning approach. We prefer an approach in which we expand the classical system by adding new metrics using molecular data. 

---

Reviewer #3: Review: Analysis of 13,000 benthic invertebrate samples from German streams reveals minor deviations in ecological status class between abundance and presence/absence data

The Manuscript(MS) deals with a very interesting topic, whether presence/absence data applicable for present monitoring activity and if it gives the same results as in case of abundance based metric.

TH MS is clear, contains a lot of efforts and analysis to support it findings. The used statistical tools are adequate to answer the given scientific questions.

I have just minor comments on the MS

Title: Why not put the exact number 13,312 in the title?

Reply: Done as requested

L31 “cheap and accurate” this phrase only occurs in the abstract not mentioned in the Introduction please give a reference to why eDNA is cheaper and accurate than traditional methods.

Reply: It is a difficult point, we changed the wording here. There are papers such as Elbrecht et al. (2017) and Aylagas et al. (2018) that suggest it can be much cheaper. However, we think its not the price that should drive the shift to the method and provided a more balanced section on the price in the discussion.

L 39 “4.” not needed

Reply: Done

L 71 “The resulting score ..” – change to The resulting score the Ecological Quality Ratio (EQR) is then. …

Generally, the term EQR is used for assessment more often.

Reply: We stick to the term ESC since its more commonly used for WFD assessment in Germany.

L77 If there are intercalibrated results compared among multiple countries why not focus only on these types and these metrics which were the intercalibration? The MS is only focusing on German stream types but with the use of a common intercalibration metric, the results could achieve more broad interest.

Reply: This is a great idea but since we have only access to the german data, for now we have to stick to this data. This analysis may be adressed in a future publication.

L 150 “The total data set contained 13,401 samples obtained from monitoring sites in 1985–2013, „

The question is why to use all of this data? This assumes that the data is the same quality from 1985 to 2013, but the WFD is much younger maybe the quality could differ from years to years and the differences of presence/absence vs. abundance assessment could be interfered.

Does it also occur that in some types there is an order of magnitude of the analysed samples than in others why not standardize or limit the number of observation among types?

Why not limit the analyses just for a smaller time-frame, like the second Water Basin Management plan frame.

Reply: This is also an interesting idea. Since this is official data from German federal states we assume an overall high quality of the data, especially because macroinvertebrates were excessively monitored using similar protocols before the WFD. Limiting the dataset to a timeframe might also add seasonal biases as well as reducing the statistical power of the analysis. Therefore, we will stick to the complete dataset. 

L 154 What was the cause of errors in the calculation? Is it just mathematical or systematic for a given typology?

Reply: Taxalists provided are not sufficient to calculate the needed metrics. Therefore, ASTERICS returns an error when trying to calculate metrics using these. 

L 159 Appendix S1 was not included in the MS

Reply: Apologies, now all scripts are included

L 255 FIG 5 should be in the supplement, not in the MS. It is not informative in this form nor the min and max values could be resolved.

Reply: As stated before it shows a trend towards generally narrower distribution of datapoints after transformation, which in our opinion adds value to the overall analysis. However, we leave the final decision to the editor.

L 279 (Table S4) This is one of the most informative tables, it should be in the MS text not in the supplement (include the number of unchanged “good” sites not only the shifts)

Reply: Table 1 in the MS gives the same information but is not limited to class shifts from 2 -> 3 or 3 -> 2. Therefore, we chose to put this much more generic table into the main MS, because it shows all shifts. However, if the reviewer and editor want us to shift it to the main MS we can for sure do so.

Fig 1. branch of “excluding data-… “ not needed, use the 13,312 samples

Reply: We think it is important to keep the current this in order to have all analyses performed on the available data sets reproducible (we had more than 13,312 initially). 

Table S4 include the number of unchanged “good” sites not only the shifts

Reply: Please see comment on Table 1 in the MS. Unchanged sites can already be easily seen there.

---

## [Decision Letter · Decision Letter 1]

3 Dec 2019

Analysis of 13,312 benthic invertebrate samples from German streams reveals minor deviations in ecological status class between abundance and presence/absence data

PONE-D-19-16057R1

Dear Dr. Leese,

We are pleased to inform you that your manuscript has been judged scientifically suitable for publication and will be formally accepted for publication once it complies with all outstanding technical requirements.

With kind regards,

Fabrizio Frontalini

Academic Editor

PLOS ONE

Additional Editor Comments (optional):

Reviewers' comments:

Reviewer's Responses to Questions

**Comments to the Author**

1. If the authors have adequately addressed your comments raised in a previous round of review and you feel that this manuscript is now acceptable for publication, you may indicate that here to bypass the “Comments to the Author” section, enter your conflict of interest statement in the “Confidential to Editor” section, and submit your "Accept" recommendation.

Reviewer #1: All comments have been addressed

Reviewer #3: All comments have been addressed

2. Is the manuscript technically sound, and do the data support the conclusions?

Reviewer #1: Yes

Reviewer #3: Yes

3. Has the statistical analysis been performed appropriately and rigorously? 

Reviewer #1: Yes

Reviewer #3: Yes

4. Have the authors made all data underlying the findings in their manuscript fully available?

Reviewer #1: Yes

Reviewer #3: Yes

5. Is the manuscript presented in an intelligible fashion and written in standard English?

Reviewer #1: Yes

Reviewer #3: Yes

6. Review Comments to the Author

Reviewer #1: I thank the authors for their thorough reply to all comments submitted in the previous review round. The most pressing issues were all addressed in the revised manuscript, and I have no further comments at this point.

Reviewer #3: (No Response)

7. PLOS authors have the option to publish the peer review history of their article (what does this mean?). If published, this will include your full peer review and any attached files.

Reviewer #1: Yes: Kevin K. Beentjes

Reviewer #3: Yes: Gabor Varbiro

---

## [Editor Report · Acceptance letter]

12 Dec 2019

PONE-D-19-16057R1 

Analysis of 13,312 benthic invertebrate samples from German streams reveals minor deviations in ecological status class between abundance and presence/absence data 

Dear Dr. Leese:

I am pleased to inform you that your manuscript has been deemed suitable for publication in PLOS ONE. Congratulations! Your manuscript is now with our production department. 

With kind regards,

on behalf of

Dr. Fabrizio Frontalini 

Academic Editor

PLOS ONE